# Repression of *GhTUBB1* Reduces Plant Height in *Gossypium hirsutum*

**DOI:** 10.3390/ijms242015424

**Published:** 2023-10-21

**Authors:** Lihua Zhang, Caixia Ma, Lihua Wang, Xiaofeng Su, Jinling Huang, Hongmei Cheng, Huiming Guo

**Affiliations:** 1Biotechnology Research Institute, Chinese Academy of Agricultural Sciences, Beijing 100081, China; zhanglihuanew@126.com (L.Z.); mcx13689481960@126.com (C.M.); 15232176080@163.com (L.W.); suxiaofeng@caas.cn (X.S.); 2National Nanfan Research Institute, Chinese Academy of Agricultural Sciences, Sanya 572024, China; 3College of Agriculture, Shanxi Agricultural University, Jinzhong 030801, China; huangjl@sxau.edu.cn

**Keywords:** cotton, transcription factor, AmCBF1, *GhTUBB1*, plant height

## Abstract

The original ‘Green Revolution’ genes are associated with gibberellin deficiency. However, in some species, mutations in these genes cause pleiotropic phenotypes, preventing their application in dwarf breeding. The development of novel genotypes with reduced plant height will resolve this problem. In a previous study, we obtained two dwarf lines, L28 and L30, by introducing the *Ammopiptanthus mongolicus* (Maxim. ex Kom.) Cheng f. *C-repeat-binding factor 1* (*AmCBF1*) into the upland cotton variety R15. We found that *Gossypium hirsutum Tubulin beta-1* (*GhTUBB1*) was downregulated in L28 and L30, which suggested that this gene may have contributed to the dwarf phenotype of L28 and L30. Here, we tested this hypothesis by silencing *GhTUBB1* expression in R15 and found that decreased expression resulted in a dwarf phenotype. Interestingly, we found that repressing *AmCBF1* expression in L28 and L30 partly recovered the expression of *GhTUBB1*. Thus, *AmCBF1* expression presented a negative relationship with *GhTUBB1* expression in L28 and L30. Moreover, yeast one-hybrid and dual-luciferase assays suggest that AmCBF1 negatively regulates *GhTUBB1* expression by directly binding to C-repeat/dehydration-responsive (CRT/DRE) elements in the *GhTUBB1* promoter, potentially explaining the dwarf phenotypes of L28 and L30. This study elucidates the regulation of *GhTUBB1* expression by AmCBF1 and suggests that *GhTUBB1* may be a new target gene for breeding dwarf and compact cultivars.

## 1. Introduction

Vigorous vegetative growth during plant development may eventually lead to a serious yield reduction [1]. Balancing the relationship between vegetative and reproductive growth, for example by breeding dwarf plants, is an important trend in crop ideotype research. Dwarf varieties have a compact architecture, which can improve crop density and lodging resistance [2,3]. In addition, dwarf crops also facilitate mechanized harvesting and reduce labor intensity [4,5]. Because dwarfing is a favorable trait in most crops [6], the cultivation of dwarf plants is an important goal in crop breeding.

The ‘Green Revolution’ genes contribute to yield increases by affecting gibberellin (GA) biosynthesis or signaling pathways [1,2]. Mutations in GA-associated genes reduce straw biomass while increasing grain-yield potential [3,7]. For example, in wheat, the semidominant mutant alleles *REDUCED HEIGHT-B1b* (*Rht-B1b*) and *REDUCED HEIGHT-D1b* (*Rht-D1b*) encode truncated DELLA proteins that cannot be degraded in response to GA signaling, resulting in improved yield traits [7] and a repressive effect on plant height [8,9]. However, both alleles also result in a grain weight penalty and more susceptibility to Fusarium head blight (FHB) [6,8,9]. *Rht24b*, a dwarfing allele of *Rht24* which encodes the GA metabolic enzyme gibberellin 2-oxidase (TaGA2ox-A9) [10], has an InDel in its promoter. This mutation results in the high expression of *TaGA2ox-A9*, which decreases endogenous bioactive GA content in the stems and reduces plant height without causing a yield penalty [11]. Moreover, *Rht24b* does not change the plant’s resistance to FHB [12]. In rice, mutated *sd1* alleles produce inactive GA20 oxidase 2 (GA20ox-2), which disrupts the conversion of GA53 to GA20 in the stem and also generates a semi-dwarf phenotype [13,14,15,16]. Although the above alleles cause dwarfism by disrupting GA synthesis or signaling, many other dwarf mutant phenotypes have been reported to result from disruption of other growth hormones. For example, mutant alleles of *BRI1*, which is involved in brassinosteroid (BR) signaling, also cause a dwarf phenotype [17,18,19,20]. Plant hormones usually affect cell division and cell elongation in the internodes to regulate plant height [21,22]. Interactions between the GA and BR signaling pathways also affect plant height. Recently, GSK3/SHAGGY-like kinase (GSK3), a negative regulator in the BR signaling pathway (known as BIN2 in *Arabidopsis*), was found to phosphorylate and stabilize Rht-B1b. A gain-of-function allele of *GSK3* was found to enhance the stability of Rht-B1b and reduce plant height in wheat [23]. In addition, genes involved in other processes may affect plant height. One example is C-repeat-binding factor (CBF), also referred to as dehydration-responsive element binding factor 1 (DREB1), which plays an important role in plant cold acclimation [24]. Interestingly, *CBF* overexpression also reduces plant height by inactivating GA [25,26]. Therefore, there are many genes that can be targeted to reduce plant height.

The identification of target dwarfing genes is of particular interest in cotton, one of the most economically important textile crops. A dwarf stature can improve planting density and enable the replacement of manual with mechanical picking, both of which directly affect the yield and cost of cotton production. Traditionally, the dwarfing of cotton has been induced by chemical spraying, artificial topping and pruning [27,28,29]. Although these practices successfully change the growth stature of upland cotton and reduce plant height, they cannot change the continuous-growth trait of cotton, which has a genetic basis. Moreover, repeated pruning to maintain dwarfing is costly. In addition, the methods used for topping are affected by various external factors such as weather and drug concentration. An alternative solution is the generation of upland cotton varieties with stably inherited dwarfism. However, there are few dwarf varieties that can be used for breeding dwarf cotton varieties. The available dwarf varieties of cotton include *AS98*, in which the upregulation of *GhDREB1* confers a dwarf phenotype [30,31]. The dwarfism of the *pag1* mutant is due to overexpression of the BR catabolism gene *PAGODA1* (*PAG1*) and the resulting deficiency in endogenous bioactive BRs [32]. In the *sd^a^* mutant, abscisic acid (ABA) biosynthesis or signaling may reduce plant height [33].

Dwarf varieties of upland cotton are more suitable for mechanical picking [34], which will become a dominant trend in the future [35]. The identification and cloning of more dwarfing-related genes can provide targets for breeding dwarf varieties and promote development of the cotton industry. In a previous study, we introduced the *Ammopiptanthus mongolicus* Cheng f. transcription factor CBF1 (*AmCBF1*) into the upland cotton variety R15 and generated two dwarf lines, L28 and L30 [36]. The comparison of differentially expressed genes among L28, L30 and R15 plants revealed that a tubulin (TUB) gene, *GhTUBB1* (*Gh_D06G2276.1*), was downregulated in L28 and L30, suggesting that it may be involved in height regulation in upland cotton. To explore the role of *GhTUBB1* expression in plant height determination, we obtained plants with knockdown of *GhTUBB1* using virus-induced gene silencing (VIGS) technology. In addition, to explore the evolutionary divergence of TUB proteins between several dicot species, we identified the members of the cotton TUB family and examined their phylogenetic relationships. Furthermore, we demonstrated that AmCBF1 can directly bind to the upstream promoter of *GhTUBB1* to repress its transcription. This study provides new insight into the AmCBF1-mediated regulation of cotton plant height and suggests *GhTUBB1* as a novel locus for breeding dwarf cotton varieties.

## 2. Results

### 2.1. Identification of Cotton GhTUBB Proteins

Using the full-length GhTUBB1 (Gh_D06G2276.1) protein as a query, we performed BLASTP searches to identify TUB family members in four species of cotton, *Arabidopsis thaliana* (L.) Heynh., *Vitis vinifera* L. and *Theobroma cacao*. According to the BLASTP results, 35 *Gossypium hirsutum* L., 31 *Gossypium barbadense* L., 17 *Gossypium raimondii* L., 19 *Gossypium arboreum* L., 9 *Arabidopsis thaliana*, 11 *Vitis vinifera* and 11 *Theobroma cacao* proteins were identified as highly reliable hits with a threshold of E-value = 0 or percentage identity greater than 90% (Appendix A). To determine the phylogenetic relationships between TUBs from cotton and other plants, a phylogenetic tree was constructed using the Neighbor-Joining method with 2000 bootstrap replicates (Figure 1). The TUB proteins were divided into six groups according to the phylogenetic relationship and genetic distance (Figure 1). Orthologous clusters were also identified based on the phylogenetic tree, and most of the cotton TUBs were orthologous to TUBs in *Arabidopsis thaliana*, *Vitis vinifera* and *Theobroma cacao*. In total, Group VI (eight orthologous clusters) contains more orthologous clusters than other groups (Figure 1). GhTUBB1 (Gh_D06G2276.1) belongs to the orthologous cluster 2 of Group IV, which includes the cotton proteins Gbar_D06G000230.1, Gorai.010G003800.1, Gh_A06G0038.1, Gbar_A06G000130.1 and Ga06G0039.1, but no orthologous proteins from other species, which may suggest that this branch evolved independently in *Gossypium* (Figure 1).

To further depict the features of TUB members, we analyzed the gene structures and protein domains of *G. hirsutum* TUBs. All of the *GhTUB* genes contained three exons and two introns, except *Gh_A07G1077.1*, *Gh_D09G1534.1* and *Gh_Sca005103G03.1* (Figure 2). Interestingly, the introns of cotton *TUB* genes, both within and between clades, tend to differ in length. In particular, the *GhTUBs* of Group IV have an obviously larger first intron than the other group *GhTUBs*, which may have been generated by insertion of an extra fragment (Figure 2A,B). These differences in intron length indicate the divergence of the cotton *GhTUB* genes. All of the GhTUBs contain both a GTPase domain and C-terminal domain (Figure 2C).

### 2.2. Chromosome Distribution and Synteny of G. hirsutum TUBB1 Family Members

The cotton *TUB* genes are unevenly distributed across the 13 chromosomes in the four *Gossypium* species, with a relatively high density on chromosomes ChrA03 and ChrD03 in both *G. hirsutum* and *G. barbadense*, Chr01 and Chr05 in *G. arboreum*, and Chr03 in *G. raimondii* (Figure 3). Synteny analysis was performed to explore the species specificity of cotton *TUBs* using the MCScanX program of TBtools software [37]. The *G. arboreum* genome has 12 and 13 syntenic *TUB* gene pairs with the *G. hirsutum* and the *G. barbadense* A subgenomes, respectively (Figure 4). Species-specific synteny was found for six TUB gene pairs. Two syntenic *TUB* gene pairs are present in the *G. arboreum* genome and *G. hirsutum* A subgenome, but not found in syntenic regions of the other cotton genomes, and four syntenic *TUB* gene pairs between the *G. arboreum* genome and *G. barbadense* A subgenome have been lost in the *G. arboreum* genome and *G. hirsutum* A subgenome (Figure 4). Four *TUB* gene pairs are syntenic between the *G. hirsutum* D subgenome and *G. raimondii* genome and have been lost in the *G. barbadense* D subgenome and *G. raimondii* genome (Figure 4). The *G. hirsutum* D subgenome and *G. raimondii* genome contain more syntenic *TUB* genes than found in other pairwise comparisons. Our data suggest that there has been lineage-specific loss and gain of TUB genes.

### 2.3. Silencing of the GhTUBB1 Gene Reduces Plant Height

According to the above analysis, *GhTUBB1* (*Gh_D06G2276.1*) exhibits several interesting characteristics, such as a lack of synteny with the *G. raimondii* genome and a large first intron, which indicate that the expression of *GhTUBB1* may be regulated by the intron and that it may play an important role in growth and development specifically in *G. hirsutum*. A previous analysis revealed that *GhTUBB1* expression was lower in L28 and L30 than in R15 (Appendix A) [36]. This indicates that *GhTUBB1* may be a downstream target of AmCBF1 and contribute to the dwarf phenotype of L28 and L30. To determine the contribution of *GhTUBB1* to cotton plant height, we specifically repressed *GhTUBB1* expression using the VIGS method (Appendix A). *pTRV2::CLA1* was included as a positive control to verify the feasibility of VIGS (Figure 5A). The *GhTUBB1*-silenced plants were approximately 30% shorter than the control plants (Figure 5A,C). Moreover, the root length was also reduced by approximately 30% in *GhTUBB1*-silenced plants (Figure 5B,D). According to quantitative reverse-transcription PCR (qRT-PCR) results, the relative expression of *GhTUBB1* was more than 40% lower in the *GhTUBB1*-silenced plants than in the control (Figure 5E). These results indicate that the silencing of *GhTUBB1* in R15 causes a dwarf phenotype and suggest *GhTUBB1* as a new target locus for optimizing plant height. 

### 2.4. AmCBF1 Silencing Partially Restores Plant Height and GhTUBB1 Expression

To further confirm the effects of AmCBF1 on *GhTUBB1* expression, we silenced *AmCBF1* in L28 and L30, and determined the effects on *GhTUBB1* expression and plant height. We observed partial restoration of plant height by silencing *AmCBF1* in L28 and L30, consistent with a previous report [36] (Figure 6A,B). qRT-PCR analysis revealed that the expression of *GhTUBB1* was also partially recovered significantly higher in the *AmCBF1*-silenced L28 and L30 plants compared with the no-VIGS control (Figure 6C). Thus, *GhTUBB1* expression is negatively correlated with *AmCBF1* expression in L28 and L30.

### 2.5. AmCBF1 Binds Directly to the GhTUBB1 Promoter

Both *AmCBF1* overexpression and *GhTUBB1* silencing reduced plant height. *GhTUBB1* transcription was also lower in *AmCBF1*-overexpression lines. These results indicate that AmCBF1 and GhTUBB1 may be involved in the same pathway controlling plant height in L28 and L30. To determine whether AmCBF1 binds the *GhTUBB1* promoter, we examined the 3.0 kb upstream of six group IV *GhTUB* genes with a large first intron. The light-responsive elements were the most abundant elements, and numerous MYB- and MYC-binding *cis*-acting elements were found in these promoters (Figure 7). This suggests that the expression of these *GhTUB* genes is responsive to the combinational regulation of light, MYB and MYC. In addition, several elements responsive to phytohormones, stress and defense were also found upstream of these *GhTUB* genes (Figure 7). The *Gh_D06G2677* promoter contains three CRT/ DRE *cis*-elements (Figure 7), which are the binding sites of CBF transcriptional factors. We examined the ability of AmCBF1 to bind to the CRT/DRE elements in the *GhTUBB1* promoter by performing a yeast one-hybrid (Y1H) assay. The relative locations of three CRT/DRE *cis*-elements in the *GhTUBB1* promoter and the structures of relevant vectors are shown in Figure 8A. The yeast strains expressing *pGADT7-AD* and *pAbAi-GhTUBB1 pro* were inviable on SD/− Leu/−Ura medium supplemented with AbA (500 ng/mL), whereas those expressing *pGADT7-AmCBF1* and *pAbAi-GhTUBB1* survived well under the same conditions (Figure 8B). This result suggested that AmCBF1 could directly bind to the *GhTUBB1* promoter, which contained the CRT/DRE elements.

To verify that AmCBF1 regulated *GhTUBB1* expression, we also performed a dual-luciferase assay in tobacco leaves (Figure 8C). The tobacco leaves injected with an *Agrobacterium* suspension containing the reporter plasmid *GhTUBB1 pro::LUC* and effector plasmid *35S::AmCBF1-GFP* showed significantly reduced LUC luminescence intensity and relative LUC/Renilla luciferase (REN) activity (approximately 30% lower) compared with the control (Figure 8D,E). Taken together, these results suggest that AmCBF1 can directly bind to the *GhTUBB1* promoter and repress its expression.

## 3. Discussion

TUBs are a medium-sized gene family in *A. thaliana*, *V. vinifera*, *T. cacao* and cotton (Figure 1 and Appendix A). In the diploids *G. raimondii* and *G. arboreum*, the number of *TUB* genes is almost double that in *A. thaliana*, *V. vinifera* and *T. cacao* (Figure 1 and Appendix A). Thus, the *TUB* genes may have undergone duplication in the *Gossypium* lineage. The number of *GhTUBs* (35 genes) is approximately equal to the sum of *GrTUBs* (17 genes) and *GaTUBs* (19 genes). However, there are fewer than 31 *GbTUBs* (31 genes) (Figure 1 and Appendix A), indicating that gene loss occurred after the tetraploidization event. The different distributions of *TUBs* on chromosomes in different species and the inconsistent synteny between the tetraploid and diploid species indicate that TUB genes have diverged between cotton species (Figure 3 and Figure 4).

In the past decades, studies of the regulation of microtubule assembly and depolymerization have mainly focused on microtubule-associated proteins (MAPs), which can regulate microtubule dynamics by directly binding to microtubule lattices [38]. Previous studies demonstrated that the MAP-mediated disruption of microtubule dynamics affected cell division and elongation [39]. ‘Green Revolution’ genes have been widely applied in the breeding of rice and wheat [6,14], but the pleiotropic phenotypes of GA-deficient mutants in other species may reduce yield [40,41]. Recently, a QWRF family member, named Reducing Plant Height 1 (ZmRPH1), was identified as a novel MAP protein regulating the cortical microtubule orientation in maize. The overexpression of *ZmRPH1* in maize reduces plant height by repressing the internode cell elongation without significantly reducing yield [42], suggesting the regulation of microtubule dynamics to restrict cell elongation as a practical strategy for breeding dwarf and compact cultivars.

Several MAPs and microtubule proteins have been reported to be involved in cotton fiber development, such as GhMAP20L5, GhTUB1 and GhTUA9 [43,44,45]. However, the specific functions of MAPs and microtubules in cotton plant height regulation remain unclear. In cotton, a dwarf and compact morphology is advantageous for mechanical harvesting, which lowers labor costs [34,35], except for the lodging resistance and photosynthetic efficiency [2,3,46]. In this study, we demonstrated that the silencing of *GhTUBB1* in the R15 variety reduced plant height. *GhTUBB1* is directly downstream of AmCBF1 and its expression is positively associated with plant height in L28 and L30. This indicates that *GhTUBB1* is a new practical potential target for breeding dwarf and compact cultivars.

The CBFs belong to the APETALA 2/ETHYLENE-RESPONSIVE FACTOR (AP2/ERF) superfamily of transcription factor with a conserved AP2 domain. CBFs bind to DRE/CRT elements to regulate plant growth and development [24]. This study confirms the direct transcriptional repression of *GhTUBB1* by a CBF-like transcription factor, which results in dwarfing in upland cotton (Figure 8). In a previous study, we demonstrated that AmCBF1 repressed the expression of *GhPP2C1* and *GhPP2C2* in L28 and L30 [36]. PP2C family members are important negative regulators of abscisic acid (ABA) signaling, and the repression or silencing of *GhPP2C1* and *GhPP2C2* may activate ABA signaling and inhibit plant growth [36]. The *GhTUBB1* promoter also contains GA-responsive *cis*-elements (Figure 7), suggesting that it may be regulated by GA-responsive transcription factors. Because GA and ABA signaling usually have antagonistic effects in plants [47,48,49] and silencing of *GhPP2C1*, *GhPP2C2* and *GhTUBB1* causes similar phenotypes in cotton, it will be interesting to explore the synergistic effect or additive effect between *GhPP2Cs* and *GhTUBB1* in controlling plant height.

## 4. Materials and Methods

### 4.1. Identification of TUB Family Members in Cotton

The full-length GhTUBB1 (Gh_D06G2276.1) protein was used as a query in BLASTP searches of the databases of four cotton species, *A. thaliana*, *V. vinifera* and *T. cacao*. The hits with an E-value = 0 or a percentage identity greater than 90% were kept. The *Arabidopsis* genome was downloaded from The Arabidopsis Information Resource (TAIR, https://www.arabidopsis.org, accessed on 28 May 2023). The *V. vinifera* genome (Genome assembly: PN40024.v4) was downloaded from Ensemble plant (http://plants.ensembl.org/index.html, accessed on 29 May 2023) [50]. The *T. cacao* genome (Cocoa Criollo B97-61/B2 version 2) was downloaded from Cocoa Genome Hub (https://cocoa-genome-hub.southgreen.fr/node/4, accessed on 29 May 2023) [51]. The genomes of *G. hirsutum* (AD, NAU assembly) [52], *G. barbadense* (AD, HAU assembly) [53], *G. arboreum* (A, CRI assembly) [54], and *G. raimondii* (D, JGI assembly) [55] were downloaded from the Cotton Functional Genomics Database (CottonFGD, https://cottonfgd.net, accessed on 28 May 2023) [56]. 

### 4.2. Phylogenetic Analysis

The amino acid sequences of the TUBs were aligned using Clustal W in MEGA 11 [57]. The full-length TUB protein sequences of cotton, *A. thaliana*, *V. vinifera* and *T. cacao* were used for phylogenetic analysis using MEGA 11 [57]. The TUB sequences in this study were highly homologous (*p*-distance was 0.07; identity = 0.93), which indicated the small evolutionary distances of TUBs and were appropriate for using the Neighbor-Joining method. Therefore, a phylogenetic tree was constructed using the Neighbor-Joining method with 2000 bootstrap replicates [57]. The phylogenetic tree was displayed using iTOL (https://itol.embl.de/, accessed on 7 June 2023) [58].

### 4.3. Gene Structure, Conserved Domain and Promoter Analysis

The exon-intron structures of *GhTUB* genes were extracted from the genome GFF annotation file and depicted using TBtools [37]. The conserved domains were generated using the Conserved Domain Database (CDD) in NCBI (https://www.ncbi.nlm.nih.gov/, accessed on 31 May 2023). Finally, the gene structure, conserved domains and phylogenetic tree of GhTUBs were visualized using Gene Structure View (Advanced) in TBtools [37]. The upstream 3.0 kb promoter regions of *GhTUBs* were submitted to the PlantCare database (http://bioinformatics.psb.ugent.be/webtools/plantcare/html/) to identify the potential *cis*-elements [59]. The distribution of *cis*-elements in the *GhTUB* promoters was visualized using TBtools [37].

### 4.4. Chromosomal Mapping and Gene Syntenic Analysis

The chromosomal locations of the cotton *TUB* genes were determined using TBtools [37] according to the cotton genomic DNA sequence GFF annotation files (CottonFGD, https://cottonfgd.net, accessed on 28 May 2023) [56]. The syntenic relationships of the cotton *TUB* genes were analyzed using One-Step MCScanX-SuperFast software in TBtools [37] according to the corresponding cotton genome annotation file (CottonFGD, https://cottonfgd.net, accessed on 28 May 2023) [56].

### 4.5. Plant Materials and Growing Conditions

The *AmCBF1* transgenic lines of L28 and L30 were generated in our previous study by introducing the full-length coding sequence of *AmCBF1* of *Ammopiptanthus mongolicus* into the cotton variety R15 [36]. The plants of cotton variety R15, the *AmCBF1* transgenic lines of L28 and L30, and tobacco (*Nicotiana benthamiana*) were grown in a greenhouse under the following conditions: 25 ± 3°C, 50% relative humidity, and a 16 h light/8 h dark cycle.

### 4.6. Quantitative Real-Time PCR

Total RNA was extracted from cotton leaves and reverse transcribed into first-strand cDNA for quantitative PCR (qPCR) using HiScript III RT SuperMix (Vazyme Biotech, R323-01, Nanjing, China). The qPCR reaction mixture was run on an ABI Prism 7500 Fast Real-Time PCR System (Applied Biosystems, Carlsbad, CA, USA) in a 20 μL reaction containing 10 μL ChamQ Universal SYBR qPCR Master Mix (Q711-02/03, Vazyme Biotech, China), 0.4 μL 10 μM primers, 200 ng cDNA and nuclease-free water. The PCR program was as follows: 95 °C (30 s), followed by 40 cycles at 95 °C (10 s), 60 °C (30 s) and 72 °C (15 s). The relative expression levels were calculated using the 2^−ΔΔCt^ method. For all qPCR assays, the cotton *UBQ* gene (*Gh_D13G1489*) was used as an internal control. The primers used for qPCR are listed in Appendix A.

### 4.7. Virus-Induced Gene Silencing

VIGS has been widely used to silence target genes in various plant species [60]. We silenced the *GhTUBB1* and *AmCBF1* genes by following a previously reported protocol. In brief, the specific gene fragment was inserted into the *pTRV2* vector via the *Xba I* and *Kpn I* restriction sites. Then, the constructed plasmids were transferred into the *Agrobacterium* strain *GV3101*. The positive-transformed strains were incubated overnight in liquid Luria-Burtani (LB) medium at 28 °C and 200 rpm. Cells were resuspended to a density of OD600 = 1.2 with permeate solution (10 mM MgCl_2_, 10 mM MES [pH 5.6], and 20 μM acetosyringone) and incubated for 3 h at room temperature in a dark chamber. Then, the suspension containing the *pTRV1* plasmid was mixed with the suspension containing one of the above *pTRV2* plasmids at a volume ratio of 1:1. The cell suspensions containing *pTRV2* and *pTRV2-CLA1* plasmids mixed with the *pTRV1* suspension were regarded as the negative and positive controls, respectively. The mixed suspensions were injected into cotton seedlings on the abaxial surface of the expanded cotyledons. The infected plants were cultured for 24 h in a dark chamber, and then grown for 4 weeks in a 23 °C chamber with a photoperiod of 16 h light/8 h dark and 50% relative humidity. The primers used for VIGS vector construction and qRT-PCR detection are listed in Appendix A.

### 4.8. Yeast One-Hybrid Assay

Three CRT/DRE elements were identified in the 3 kb promoter region upstream of the *GhTUBB1* gene. A Y1H analysis was conducted to verify the binding of AmCBF1 to the *GhTUBB1* promoter. The full-length *AmCBF1* coding sequence was inserted into the *pGADT7* vector via the *EcoR I* and *BamH I* restriction sites to construct the prey plasmid *pGADT7-AmCBF1-AD*. The *GhTUBB1* promoter containing all CRT/DRE elements was inserted into the *pAbAi* vector via the *Kpn I* and *Sal I* restriction sites to generate the bait plasmid *pAbAi-GhTUBB1 pro*. Then, the bait plasmid was linearized by *Bstb I* digestion and introduced into the Y1H Gold yeast strain to generate a decoy reporter yeast strain. An aureobasidin A (AbA) inhibition assay was used to select the positive strains. The prey plasmid was then transformed into the bait reporter strains to test DNA–protein interactions. The yeast cells containing the prey and bait plasmids were cultured on SD/−Leu/−Ura medium for 3 days at 30 °C. Subsequently, these yeast cultures were diluted (1:10, 1:100 and 1:1,000) and cultured on SD/−Leu/−Ura media supplemented with different concentrations of AbA. The *pGADT7* vector was used as the negative control. The primers used in this experiment are listed in Appendix A.

### 4.9. Dual-Luciferase Assay

The effector plasmid *pCambia1302-35S::AmCBF* was constructed by inserting the full-length coding sequence of *AmCBF1* into the *pCambia1302* vector via the *Nco I* restricted site. The *GhTUBB1* promoter containing three CRT/ DRE elements was inserted into the *pGreenII 0800-LUC* vector to generate the reporter plasmid *pGreenII 0800-GhTUBB1 pro::LUC*. Then, the recombinant plasmids and the negative control plasmid were introduced into *Agrobacterium* strain *EHA105*. The positive strains were cultured to OD_600_ = 1.0 at 28 °C and 200 rpm. The strains were resuspended to OD_600_ = 1.2 with permeate solution (10 mM MgCl_2_, 10 mM MES [pH 5.6], and 150 μM acetosyringone). The suspensions were incubated for 3 h at room temperature in the dark. Then, the suspensions containing effector and reporter plasmids were mixed and injected into tobacco leaves. After 48–60 h of growth in a chamber at 23 °C with a photoperiod of 16 h light/8 h dark and 50% relative humidity, the leaves were sprayed with 10 μM D-Luciferin potassium salt and then imaged using a LB985 Night SHADE fluorescence imaging system (Berthold Technologies, Bad Wildbad, Germany). The dual LUC activity was measured using a GloMax 20/20 luminometer (Promega, Madison, WI, United States), and relative activity was calculated according to the Dual-Luciferase Reporter Assay System instructions (Promega). The primers used in this experiment are listed in Appendix A.

## 5. Conclusions

*GhTUBB1* was identified as being differentially expressed in L28 and L30 vs. R15 according to previous RNA-sequencing data. The silencing of *GhTUBB1* expression caused a dwarf phenotype in cotton plants. *GhTUBB1* expression was negatively associated with *AmCBF1* expression in L28 and L30. Moreover, AmCBF1 directly bound to CRT/DRE elements and negatively regulated the expression of *GhTUBB1*, which may explain the dwarf morphology of L28 and L30 (Figure 8F). This study elucidated the regulation of *GhTUBB1* expression by AmCBF1 and suggested *GhTUBB1* as a new potential target gene for breeding dwarf and compact cultivars.

## Figures and Tables

**Figure 1 ijms-24-15424-f001:**
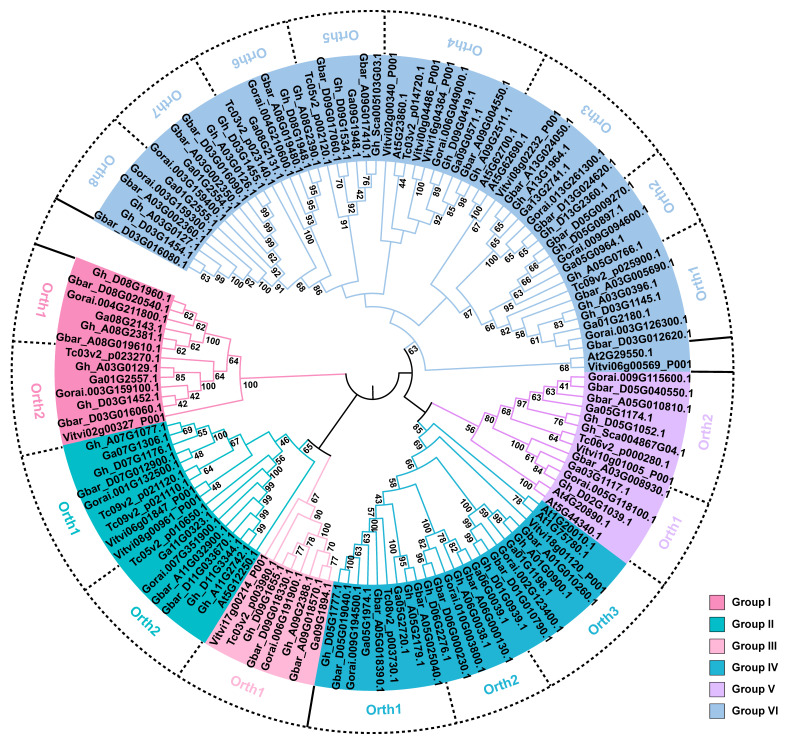
Phylogenetic tree of the TUB proteins from seven plant species generated using the Neighbor-Joining method. Colors indicate the six groups of TUB proteins. Gh: *Gossypium hirsutum*, Gbar: *Gossypium barbadense*, Ga: *Gossypium arboretum*, Gorai: *Gossypium raimondii*, At: *Arabidopsis thaliana*, Vitvi: *Vitis vinifera*, Tc: *Theobroma cacao*. Orth indicates the potential groups of orthologous genes.

**Figure 2 ijms-24-15424-f002:**
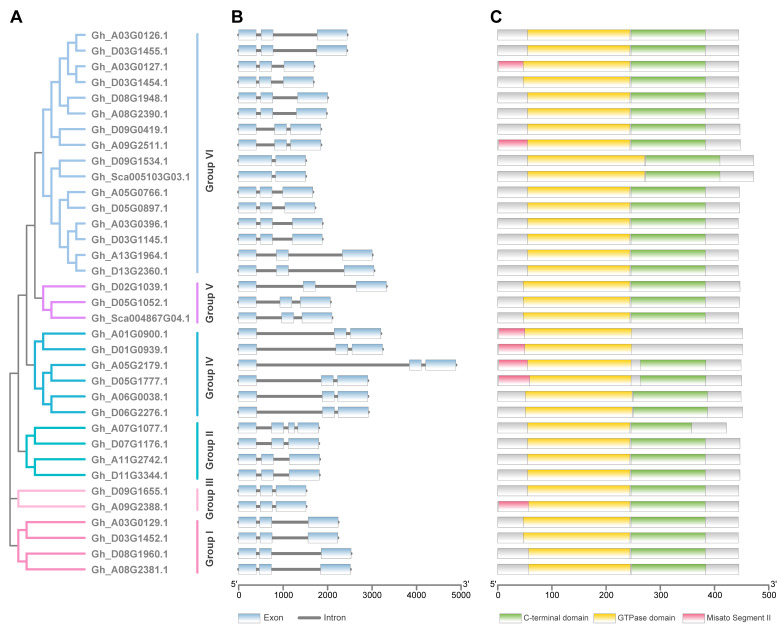
Phylogenetic relationship conserved motifs of GhTUB proteins and structures of *GhTUB* genes. (**A**) Phylogenetic tree of GhTUB proteins in *G. hirsutum*. (**B**) Exon-intron structure of *GhTUB* genes. (**C**) Distributions of three conserved domains in GhTUB proteins.

**Figure 3 ijms-24-15424-f003:**
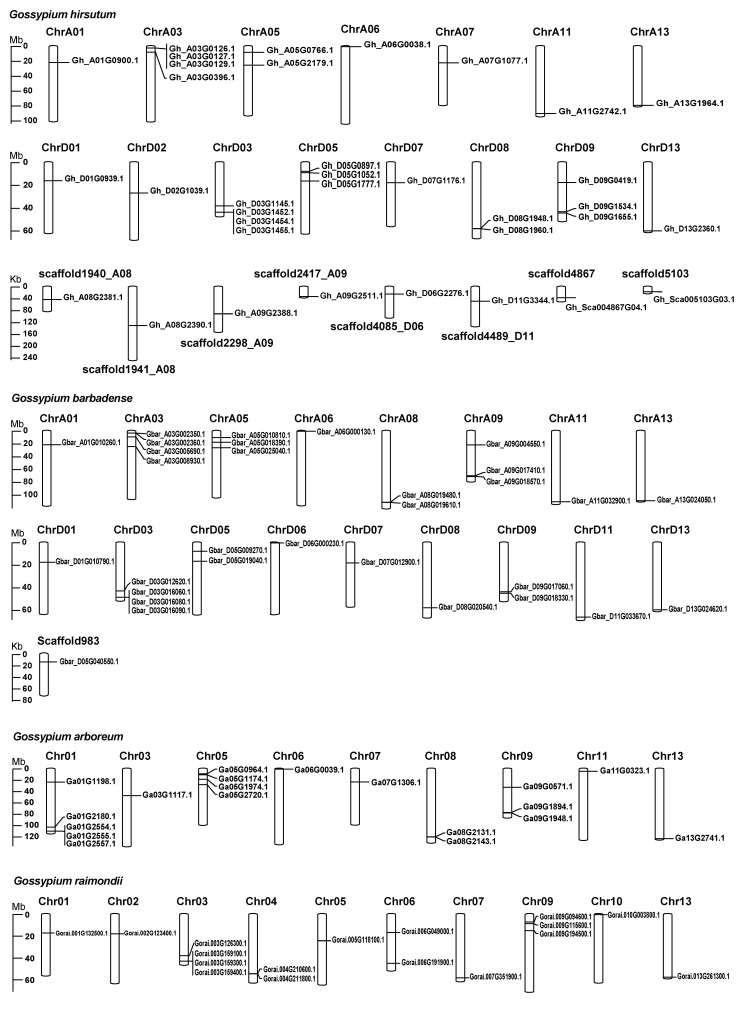
Chromosomal localization of *TUB* genes in different cotton species.

**Figure 4 ijms-24-15424-f004:**
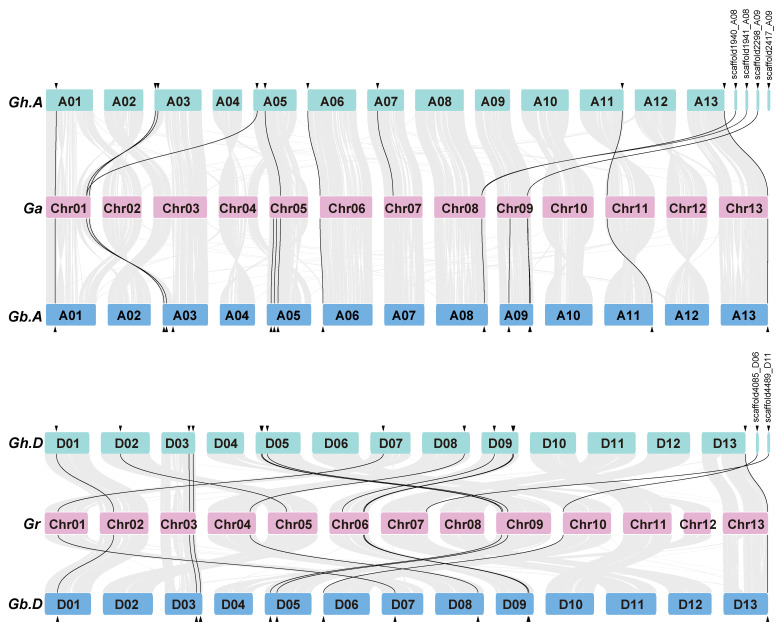
Synteny of *TUB* genes among the diploid and allotetraploid cotton species. Gray lines indicate collinear blocks within the cotton genomes, and black lines represent the syntenic *TUB* gene pairs. Chromosomes of the representative cotton species are shown in different colors. Gh.A, *G. hirsutum* A subgenome; Ga, *G. arboreum* genome; Gb.A, *G. barbadense* A subgenome; Gh.D, *G. hirsutum* D subgenome; Gr, *G. raimondii* genome; and Gb.D, *G. barbadense* D subgenome..

**Figure 5 ijms-24-15424-f005:**
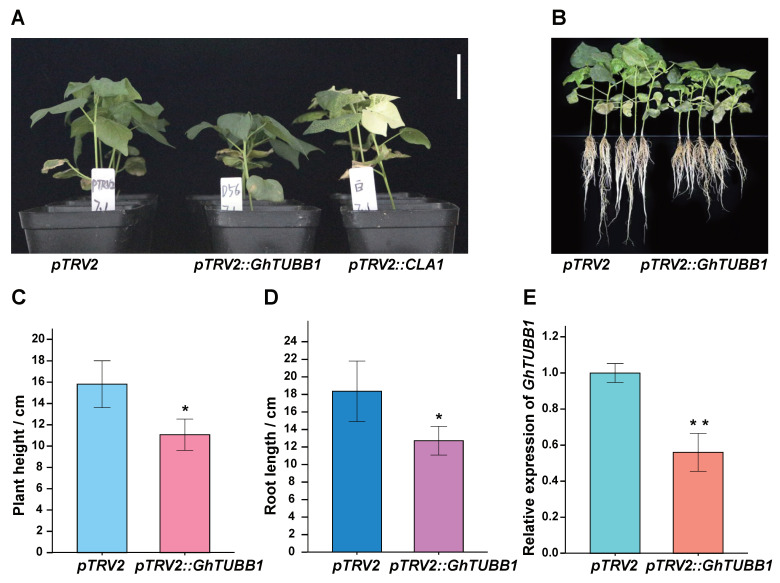
Preliminary analysis of *GhTUBB1* gene functions. (**A**,**B**) Silencing of *GhTUBB1* in R15 reduced the plant height (**A**) and root length (**B**). VIGS using the *pTRV2::CLA1* plasmid, which causes an albino phenotype, was performed as a positive control. *pTRV2* indicates VIGS performed with the empty *pTRV2* vector as a negative control. Bar is 4 cm. (**C**–**E**) Quantification of plant height (**C**), root length (**D**), and relative gene expression (**E**) after silencing of *GhTUBB1* in R15. Error bars indicate the standard error. * *p* < 0.05, ** *p* < 0.01 (Student’s *t*-test).

**Figure 6 ijms-24-15424-f006:**
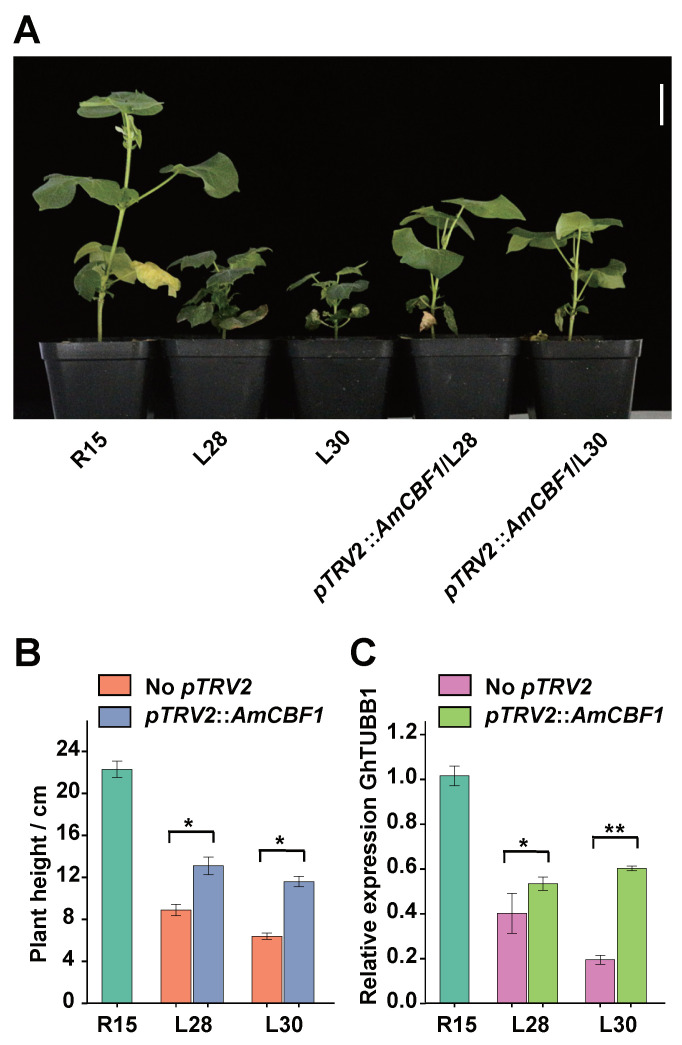
*AmCBF1* silencing partially restored *GhTUBB1* expression in L28 and L30. (**A**) *AmCBF1* silencing partially restored the height of L28 and L30. Bar is 4 cm. (**B**) *AmCBF1* silencing significantly increased the height of L28 and L30. (**C**) *GhTUBB1* expression significantly increased after *AmCBF1* silencing in L28 and L30. Error bars indicate the standard error. * *p* < 0.05, ** *p* < 0.01 (Student’s *t*-test). No *pTRV2* indicates the negative control without VIGS.

**Figure 7 ijms-24-15424-f007:**
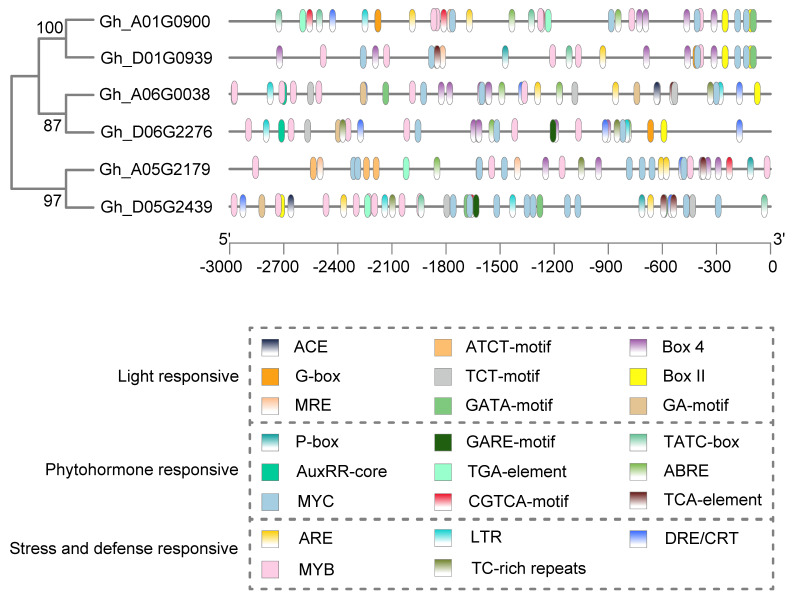
Analysis of *cis*-elements in six *GhTUB* promoters. Boxes filled with different colors represent different *cis*-elements.

**Figure 8 ijms-24-15424-f008:**
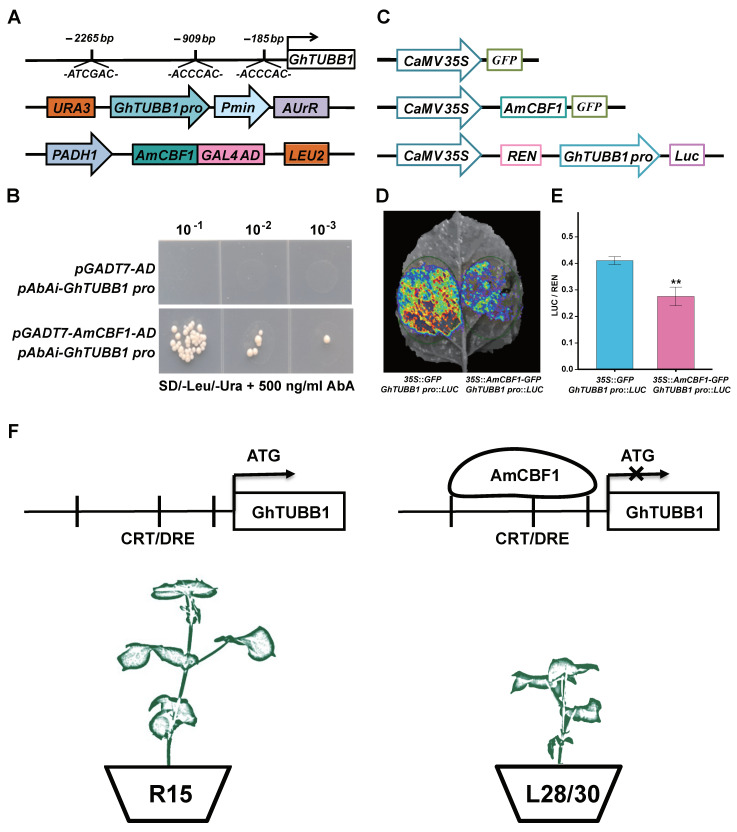
AmCBF1 represses the expression of *GhTUBB1*. (**A**) Schematic diagram of the CRT/DRE binding elements in the *GhTUBB1* promoter and the plasmids used for yeast one-hybrid assays. (**B**) AmCBF1 binds to the *GhTUBB1* promoter according to a yeast one-hybrid assay. Yeast cells were grown at serial dilutions of 1:10, 1:100 and 1:1,000 on SD/−Leu/−Ura medium supplemented with 500 ng mL^−1^ aureobasidin A (AbA). (**C**) Schematic diagram of the reporter and effector plasmids used for transient transcriptional activity assays in tobacco. REN, Renilla luciferase; LUC, firefly luciferase. (**D**) AmCBF1 repressed the fluorescence signal activated by *GhTUBB1* promoter activation in a dual-luciferase assay. Red indicates the strongest fluorescence signal intensity. (**E**) *GhTUBB1* expression decreased significantly upon *AmCBF1* expression. *35S::GFP* represents the empty vector *pCambia1302-35S::GFP*, which was used as the negative control. LUC activity was normalized to REN activity and expressed as relative expression. Error bars indicate the standard error. ** *p* < 0.01 (Student’s *t*-test). (**F**) Diagram of a working model of AmCBF1 regulation of the *GhTUBB1* gene. Transcription factor AmCBF1 inhibits the expression of *GhTUBB1* and negatively regulates the growth and development of upland cotton seedlings by binding to CRT/DRE elements.

## Data Availability

The *Arabidopsis* genome was downloaded from The Arabidopsis Information Resource (TAIR, https://www.arabidopsis.org). The *V. vinifera* genome (Genome assembly: PN40024.v4) was downloaded from Ensemble plant (http://plants.ensembl.org/index.html) [50]. The *T. cacao* genome (Cocoa Criollo B97-61/B2 version 2) was downloaded from Cocoa Genome Hub (https://cocoa-genome-hub.southgreen.fr/node/4) [51]. The genomes of *G. hirsutum* (AD, NAU assembly) [52], *G. barbadense* (AD, HAU assembly) [53], *G. arboreum* (A, CRI assembly) [54], and *G. raimondii* (D, JGI assembly) [55] were downloaded from the Cotton Functional Genomics Database (CottonFGD, https://cottonfgd.net) [56]. The cotton genomic DNA sequence GFF annotation files was downloaded from the Cotton Functional Genomics Database (CottonFGD, https://cottonfgd.net) [56].

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
