# Peer review of "Repression of *GhTUBB1* Reduces Plant Height in *Gossypium hirsutum"

_ijms, 2023, doi:10.3390/ijms242015424_

Round 1

Reviewer 1 Report

Review of the article "The repression of GhTUBB1 reduce plant height in Gossypium hirsutum"

In my opinion, the article is interesting and valuable.

My points to improve the article:

1. In the Abstract is: "Thus, the GhTUBB1 expression presented negative relationship with AmCBF1 expression in L28 and L30."

Because in this sentence reason and result has been changed, in my opinion, in more logical way it should be written:

"Thus, the AmCBF1 expression presented negative relationship with GhTUBB1 expression in L28 and L30.", i.e. AmCBF1 expression as a reason and GhTUBB1 expression as a result. I am not sure whether the mechanisms you discovered works in opposite side, i.e. whether stimulation/inhibition of expression of GhTUBB1 affects AmCBF1 expression.

2. In the Results section is: "Acoording to the preliminary alignment results, there are ..." - I do not understand what "the preliminary alignment results" means, please add short explanation.

Moreover, I do not understand why you used the PRALINE multiple sequence alignment. In the MEGA program you used to generate phylogenetic trees, are "build in" programs to make alignment, especially ClustalW is well known. Is it possible to compare the results you obtained with the results that would be obtained using ClustalW implemented in the MEGA program?

3. In the Results section is: "To verify the phylogenetic relationship of TUBs between cotton and other plants, the phylogenetic tree was constructed by using the NJ method with 2200 bootstrap replicates (Figure 1)."

a) The results (i.e. the calculated phylogenetic relationship) depend on the used method and the number of replications in the bootstrap method. Please explanin in the article, why you used the NJ method. Using a different method, for example the Maximum Likelihood method, can lead to different results and conclusions.

b) The number of replication in the bootstrap method is in my opinion "strange", i.e. I do not understand why you have set 2200 replications. It is quite possible that for example for 3000 replications the results will be different. In the MEGA 11 program you used to generate phylogenetic trees, it is possible to set maximally 10000 replications in the bootstrap method. Maybe you should use this maximum possible (10000) number of replications?

c) The node reliabilities shown in Figure 1 are not very high, for this reason please discuss briefly why we can trust these results.

4. The Discussion section looks like part of Introduction. The Discussion section should be improved to focus more on the discussion of the results obtained by the Authors and presented in the article (in relation to the results of other authors).

5. The article is well written, only small language corrections need to be made, for example:

a)"Neighbor-joining" (lines 131-132) should be corrected to "Neighbor-Joining".

b) "Acoording to the preliminary" (line 98) should be corrected to "According ...".

The article is well written, only small language corrections need to be made.

Reviewer 2 Report

1). Manuscript ID: IJMS-2643616

2). Manuscript title: The repression of GhTUBB1 reduces plant height in Gossypium hirsutum

General Comments:

*Please follow the journal format, while revising the manuscript.

*Add scientific authority at the end of binomial names of all species, when they are mentioned for the first time in the manuscript.

*Include full forms of all the abbreviations/acronyms mentioned in the manuscript.

Specific Comments:

*Thoroughly proofread the manuscript before submission. English corrections required in several sections of the manuscript. Some of them are mentioned below.

*Line 2: "reduces Plant height."

*Line 16: Rewrite

*Line 70: Change to "in addition"

*Line 126: Change to "which may have been generated."

*Line 142: Change to "relatively high density"

*Line 147 and 148: Rewrite

*Line 196: Change to "was also partially recovered"

*Line 215: Rewrite

*Line 221: "are shown in Figure 8A"

*Line 355: "Were submitted to the PlantCare database"

*Line 378: Degrees C

*Line 385: "Was shown"

*Line 390: Change to "acetosyringone"

*Line 425 and 428: Rewrite

NOTE:

Please kindly see attached file for additional remarks.

*Thoroughly proofread the manuscript before submission. Extensive English corrections required in several sections of the manuscript.

Round 2

Reviewer 1 Report

The Authors have correctly addressed all my concerns and comments. Now, the article is better and in my opinion it can be published in International Journal of Molecular Sciences.